# Electromechanical Dynamics Model of Ultrasonic Transducer in Ultrasonic Machining Based on Equivalent Circuit Approach

**DOI:** 10.3390/s19061405

**Published:** 2019-03-21

**Authors:** Jian-Guo Zhang, Zhi-Li Long, Wen-Ju Ma, Guang-Hao Hu, Yang-Min Li

**Affiliations:** 1Harbin Institute of Technology Shenzhen, Shenzhen 518055, China; zhangjianguohit@foxmail.com; 2Key Laboratory of Precision Microelectronic Manufacturing Technology & Equipment of Ministry of Education of Guangdong University of Technology, Guangzhou 510006, China; 3Shenzhen Engineering Lab for Medical Intelligent Wireless Ultrasonic Imaging Technology, Shenzhen 518055, China; 4Henan University of Science and Technology, Luoyang 471003, China; m1468934516@126.com (W.-J.M.); huguanghaohenan@foxmail.com (G.-H.H.); 5The Hong Kong Polytechnic University, Kowloon, Hong Kong, China; yangmin.li@polyu.edu.hk

**Keywords:** impedance model, equivalent circuit, ultrasonic transducer, rotary ultrasonic machining (RUM)

## Abstract

Ultrasonic transducer is a piezoelectric actuator that converts AC electrical energy into ultrasonic mechanical vibration to accelerate the material removal rate of workpiece in rotary ultrasonic machining (RUM). In this study, an impedance model of the ultrasonic transducer is established by the electromechanical equivalent approach. The impedance model not only facilitates the structure design of the ultrasonic transducer, but also predicts the effects of different mechanical structural dimensions on the impedance characteristics of the ultrasonic transducer. Moreover, the effects of extension length of the machining tool and the tightening torque of the clamping nut on the impedance characteristics of the ultrasonic transducer are investigated. Finally, through experimental analysis, the impedance transfer function with external force is established to analyze the dynamic characteristics of machining process.

## 1. Introduction

Super hard materials have been applied in many fields, including medical, electronic product, industry equipment, and aeronautic technology fields [1,2,3]. Typical super hard materials include the optical glass, quartz, glass ceramics, corundum, silicon nitride, and composite materials [2,4]. Rotary ultrasonic machining (RUM) can provide the superposition of tool rotation and ultrasonic vibration on the workpiece. With the novel RUM technology, the super hard materials that are difficult to process in traditional way can be processed economically and achieve the better machining quality [4,5].

The ultrasonic vibration system of RUM consists of the power supply, contactless transformer and ultrasonic transducer. The power supply provides AC excitation current to the piezoelectric ceramic stacks, which is based on anti-piezoelectric effect to generate mechanical ultrasonic vibration. The contactless transformer is an electromagnetic induction component to transfer the AC excitation power to ultrasonic transducer through the air-gap. The ultrasonic transducer consists of piezoelectric ceramic stacks, solid horn, collet, clamping nut, and machining tool. The function of the solid horn is to enlarge the mechanical vibration, and to concentrate the ultrasonic energy on the smaller area. The machining tool transmits ultrasonic vibration to the surface of the workpiece. Many investigations have proven that the stability of ultrasonic vibration is a critical factor of ultrasonic machining [6,7,8,9,10,11]. The impedance characteristics of ultrasonic transducers depend on various factors, including the structure materials, the tightness of clamping nut, extension length of the machining tool, and machining force. Therefore, the ultrasonic transducer needs the precise impedance model to describe the ultrasonic transducer and obtain optimized ultrasonic vibration. It is significant to investigate the reliable impedance equivalent model to predict and control the machining process in RUM.

The structural design and dynamics analysis of the ultrasonic transducer have aroused general concern by many researchers and manufacturers. Currently, the modeling methods of the ultrasonic transducer include finite-element analysis, mass-spring-damper (MSD) system, transfer matrices, and equivalent circuits [6,7,8,9,10,11,12,13,14]. The finite element method (FEM) can attain the vibration modal and the resonant frequency of mechanical structure animatedly. It is commonly used to design and analyze ultrasonic transducers and is always applied to the analysis of dynamic characteristic and resonant frequency of the ultrasonic transducer [6,7,8,9,10,11,12]. The FEM can analyze the effect of different structures on the dynamic characteristic of the ultrasonic transducer by setting FEM nodes. Therefore, only the mechanical behavior of the ultrasonic transducer design can be accurately estimated by this method, and it cannot analyze the loss of electrical and external mechanical load. The ultrasonic transducer also can be modeled as a chain of the mass-spring-damper system. Saleem proposed a model based on Kelvin–Voigt model to describe one-dimensional contact interaction between the ultrasonic transducer and workpiece [13]. Voronina and Babitsky investigated a two-order equivalent MSD system to describe the piezoelectric transducer and presented that the contact interface is equivalent to nonlinear load. This equivalent electrical and mechanical system could explain the dynamics of machining process [14,15]. Wang et al. investigated the MSD system to analyze the dynamic characteristics of a thickness-mode piezoelectric transducer at its resonant frequency [16]. The analytical solution and the KLM and Mason’s equivalent circuit were investigated to produce identical impedance curves [17]. The equivalent circuit method of the ultrasonic transducer can precisely define the relationship between the electrical input and the mechanical vibration output under defined operating conditions, which rely on an accurate model of the interaction between the electrical and mechanical parameters of the ultrasonic transducer. To obtain an accurate model of ultrasonic transducer, many researchers have studied various equivalent circuit methods. Smyth investigated an analytical Mason equivalent circuit to describe the ultrasonic transducer and enable straightforward and wide-ranging model implementation for future ultrasonic transducer design and optimization [18]. Je et al. developed an advanced equivalent circuit model for the piezoelectric ultrasonic transducer; this model can be used to predict the effect of a piezoelectric layer on the coupling factor and efficiency of piezoelectric micromachined ultrasonic transducers [19]. Caronti et al. developed an accurate model for ultrasonic transducers [20]. Wang et al. presented the design of high-frequency ultrasonic transducers by using electromechanical equivalent method and three-dimensional (3-D) FEM to get the optimization geometric dimensions [21]. The equivalent circuit method can analyze the different mechanical loads, the resonant frequency, and the displacement of an ultrasonic transducer [22,23,24]. In the above research, the effects of tightness of clamping nut and extension length of machining tool, and the dynamical impedance model with different loads in numerical calculation were not mentioned. In the ultrasonic transducer vibration system, the clamping nut is a critical component to fix the machining tool. The fasten force of clamping nut influences the stress distribution and impedance characteristics of ultrasonic transducer. It is necessary to investigate the tightening torque effect of clamping nut. The machining tool’s extension length affects the wavelength distribution of solid horn structure. The different tools’ extension lengths need to be considered in assembling machining tool of ultrasonic transducer. Impedance and resonant frequency are key factors in the ultrasonic transducer vibration system. The contributions of this paper are: (1) We investigate the whole electromechanical characteristics of ultrasonic transducer by using the equivalent circuit to describe the lumped static impedance/admittance characteristic of ultrasonic transducer, (2) The ultrasonic piezoelectrical transducer is sensitive to the external mechanical load [25]. Therefore, the dynamical impedance transfer function of ultrasonic transducer with different loads is established by the parameter fitting method. Through this modelling method, the resonant frequency and impedance can be beneficial to the monitoring of machining process. In this paper, the impedance lumped model of ultrasonic transducer can calculate and predict the static and dynamic impedance characteristics of an ultrasonic transducer.

The paper is organized into six sections. Section 1 presents the introduction. In Section 2, the mechanical structure is introduced. In Section 3, the impedance equivalent circuit method is applied to impedance modeling of ultrasonic transducer. Section 4 establishes the experimental platform to verify the effect of clamping nut and machining tool extended length. The static impedance model is verified by the experimental measurement. In Section 5, The MSD and electrical models are utilized to describe the ultrasonic transducer vibration system, the impedance transfer function with external force is established to analyze the dynamic characteristic of machining process. Finally, the conclusions are presented in Section 6.

## 2. Mechanical Structure of Ultrasonic Transducer

Figure 1 shows the typical structure of the solid horn-type piezoelectric transducer. Generally, the ultrasonic transducer consists of eight main parts, including the inner screw bolt, the back slab, the piezoelectric ceramic stacks, the front slab, clamping nut, collet tool, solid horn, and machining tool. The piezoelectric ceramic stacks are clamped between the front slab and back slab. The determination of horn structure resonant wavelength usually integer multiple of half wavelength [26]. The ultrasonic vibration is realized by using the piezoelectric ceramics converts electrical energy into mechanical energy based on anti-piezoelectric effect [27,28]. This vibration amplitude of piezoelectric ceramics is still small, so the vibration of piezoelectric ceramic stacks is amplified by the horn structure. The machining tool is clamped to the head of solid horn with threaded connection of the clamping nut. The ultrasonic transducer is driven by the electric sinusoidal waveforms from an ultrasonic generator with resonant frequency tracking. Then ultrasonic mechanical vibration is applied to the workpiece.

## 3. Impedance Modeling

### 3.1. The piezoelectric Ceramic Stacks and Screw Bolt

The electromechanical equivalent method is an effective way to deal with ultrasonic transducer design [22,29]. In the equivalent circuit, *C*_0_ is the piezoelectric capacitance of the transducer. In detail, the mechanical force is equal to voltage, and the vibration velocity is equal to current. Therefore, the electromechanical equivalent circuit of piezoelectric stacks and screw bolt is shown in Figure 2. The electromechanical equivalent circuit of ultrasonic transducer is obtained by separating the piezoelectric material into an electrical port and a mechanical port by using an ideal electromechanical transformer. N is piezoelectric coupling factor and N < 1. The screw bolt and piezoelectric ceramic stacks are labelled as S and P, respectively. Subscripts L, M, and R denote the left, middle and right location of T-type equivalent impedance structure.

The static capacitance value of piezoelectric ceramic stacks is expressed by:(1)$$C_{0}=n\frac{d_{33}}{s_{33}^{E}}\frac{S}{t}$$
where $s_{33}^{E}$ is the elastic compliance, $d_{33}$ is the piezoelectric charge coefficient, *S* is the area of piezoelectric ring, and the area of the stack is $S=\pi (r_{2}^{2}-r_{1}^{2})$. *t* is thickness of the piezoelectric ring and *n* is the number of piezoelectric rings [22,29]. The piezoelectric coupling factor *N* is expressed as:(2)$$N=\frac{nS}{L_{P}}\frac{d_{33}}{s_{33}^{E}}$$
where the total length of the piezoelectric stacks and the screw bolt is $L_{P}=nt$.

The Equivalent impedances of screw bolt and piezoelectric stacks are expressed as:(3)$$Z_{R}^{S}=Z_{L}^{S}=jZ_{0}^{S}\tan(\frac{\tau^{S}L_{P}}{2})$$
(4)$$Z_{R}^{P}=Z_{L}^{P}=jZ_{0}^{p}\tan(\frac{\tau^{p}L_{P}}{2})$$
(5)$$Z_{M}^{S}=\frac{Z_{0}^{S}}{j\sin(\tau^{S}L_{P})}$$
(6)$$Z_{M}^{P}=\frac{Z_{0}^{P}}{j\sin(\tau^{P}L_{P})}$$
where $Z_{0}^{i}=\rho cS$ is the specific acoustic impedance which is the product of the density, velocity and area of the piezoelectric ring or the screw bolt. $\tau^{i}=\omega /c$ is the material propagation constant, and *c* is the material acoustic velocity.

### 3.2. The Solid Horn Structures

The electromechanical equivalent equations of the solid horn can be expressed as a T-type equivalent circuit [22,23,29], and the equivalent impedance expressions of the two horn structures are shown in Table 1.

### 3.3. The Tightening Torque of Clamping Nut

The wave speed in the material is determined by the stress and elongation. The relationship between the wave speed and the bolt elongation is formulated and utilized to develop a real-time ultrasonic control of the tightening process of bolted assemblies [30]. The clamping nut and collet tool are constructed into fastening the machining tool. They can be taken as a rigid constant section horn. The fasten force can impact the ultrasonic wave velocity of the threaded components.

The wave speed of material varies with fasten force. When the longitudinal ultrasonic wave propagates in the uniform material and regular shape object, it is affected by the axial stress of object [30,31]. The relational expression is as follows: (7)$$\frac{dc_{n}}{d\sigma }=\frac{\left[2l+\lambda +(\frac{\lambda +\mu }{\mu })(4m+4\lambda +10\mu )\right]}{2c_{n}\rho (3\lambda +2\mu )}$$
where λ and μ are Lamé or second-order elastic constants, l and m are Murnaghan’s third-order elastic constants, ρ is the material density, *c*_n_ is wave speed, and σ is the compressive stress.

The wave speed is determined by the tightening torque of clamping nut and the material properties. In the elastic range, the wave speed with the axial elongation Δ*l* is expressed as:(8)$$c_{n}={\left(c_{o}^{2}+\Lambda (\Delta l/L)\right)}^{\frac{1}{2}}$$

The clamping nut, collet tool and machining tool can be regarded as uniform and isotropic material column. The equivalent impedance expressions are:
(9)$$\left\{ \begin{array}{l} Z_{R}^{C}=Z_{L}^{C}=\rho_{c}c_{n}S_{c}(\frac{1}{j\tan(\tau_{c}L_{c})}-\frac{1}{j\sin(\tau_{c}L_{c})})\\ Z_{M}^{C}=\frac{\rho_{c}c_{n}S_{c}}{j\sin(\tau_{c}L_{c})} \end{array}\right.$$

### 3.4. Impedance Equivalent Modeling

From individual equivalent circuit and their corresponding impedance, the whole equivalent circuit for the ultrasonic transducer is integrated as Figure 3. The input impedance of the back slab, front slab, clamping nut, and machining tool which are constant for their cross-section area, are defined as constant horn. The input impedance of the solid horn is defined as the exponential horn.

Based on the Table 1, the equivalent impedance of the front slab (1) is:(10)$$Z_{F(1)}=Z_{L}^{F}+\frac{Z_{M}^{F}\times {(Z}_{R}^{F}{+Z}_{F(2)})}{Z_{M}^{F}{+(Z}_{R}^{F}{+Z}_{F(2)})}$$

Similarly, the input equivalent impedance Z_C_, Z_H_, Z_F(2)_ in Figure 3 can be expressed in the same formula form. Then, the equivalent impedance of right side of location (a) in Figure 3 is shorted as:(11)$$Z_{\mathrm{right}}{=Z}_{R}^{P}{+Z}_{R}^{S}{+Z}_{F(1)}$$

Similarly, the equivalent impedance of left side of location (a) in Figure 3 is:(12)$$Z_{\mathrm{left}}{=Z}_{L}^{P}{+Z}_{L}^{S}{+Z}_{R}^{B}+\frac{Z_{L}^{B}\times Z_{M}^{B}}{Z_{L}^{B}+Z_{M}^{B}}$$

Therefore, the total mechanical impedance of the horn is:(13)$$Z_{m}{=Z}_{M}^{S}{+Z}_{M}^{P}+\frac{Z_{\mathrm{right}}\times Z_{\mathrm{left}}}{Z_{\mathrm{right}}{+Z}_{\mathrm{left}}}$$

The electromechanical equivalent impedance of the ultrasonic transducer is expressed as:(14)$$Z_{e}=\frac{1}{Y_{e}}=\frac{U}{I}=\frac{Z_{C_{0}}\times N^{2}Z_{m}}{Z_{C_{0}}+N^{2}Z_{m}}$$
where $Z_{C_{0}}=1/j\omega C_{0}$, *Y*_e_ is admittance, and *N* is the coupling factor defined in Equation (2).

Therefore, the electrical impedance model of the ultrasonic transducer vibration system is established as Equation (14). This lumped equivalent impedance/admittance model includes all parameters, including wave speed and the density of material, the mechanical and electrical losses in material and the structural dimensions.

## 4. Numeral Calculation and Discussion

From the established impedance model, the frequency and impedance of the ultrasonic transducer at different loads are calculated in MATLAB. The properties of the ultrasonic transducer are listed in Table 2, and the material loss can be defined as imaginary part in the elastic module [29]. In experiment, the impedance of the ultrasonic transducer is measured frequency sweeping by using an Agilent 4294A impedance analyzer. The axial force testing platform is established by the Z-axis motor motion structure, the motor driver (model ATK-2MD4850), the force sensor (model FB10-100 kg), and the data acquisition device (model JL-DT01). This platform can realize the identification of dynamic impedance model of ultrasonic transducer with axial force. The experimental platform is shown in Figure 4.

### 4.1. Load of Ultrasonic Transducer

When the ultrasonic transducer is loaded with the machining tool, the parameters of materials and structural dimensions based on Table 2 are substituted in Equation (14). The impedance characteristics of ultrasonic transducer are shown in Figure 5. It is found that the corresponding conductance, susceptance and resonant frequency get closed to the experimental measurement results of ultrasonic transducer. When the conductance is at maximum value and the susceptance is zero, the phase of ultrasonic transducer is zero and corresponding impedance of ultrasonic transducer is lowest, the current frequency is the resonant frequency of ultrasonic transducer. When the ultrasonic transducer operates at resonant frequency, the active power of ultrasonic transducer reaches maximum. It means that the electrical equivalent model can accurately describe the ultrasonic transducer in RUM.

### 4.2. Load with Different Torques of Clamping Nut

The resonant frequency and impedance characteristics of the ultrasonic transducer with different torques of clamping nut are obtained as shown in Figure 6. The experimental data validates the sound velocity effect on the trend of resonant frequency. It is observed that when the fasten force increases with the tightening torque, the sound velocity increases with the torque. When the tightening torque becomes larger, the resonant frequency of ultrasonic transducer increases, while the corresponding impedance of ultrasonic transducer at the resonant frequency decreases.

### 4.3. Load with Different Extension Lengths of Tool

When the ultrasonic transducer is in operation, the machining tool wears down due to heat and friction, resulting in poor processing quality of processed object. It is necessary to replace the broken machining tool regularly. In the experiment, the admittance circle and the resonant frequency of ultrasonic transducer are measured by the impedance analyzer. The measurement results are shown in Figure 7. The admittance circle is changed with different extension lengths of machining tool in Figure 7a. It is found that the extension length of machining tool strongly influences the vibration characteristics of ultrasonic transducer. When the extension length of the machining tool becomes longer, the resonant frequency of the ultrasonic transducer decreases (Figure 7b).

## 5. Dynamic Modeling of Ultrasonic Transducer

When the ultrasonic transducer is in the no-load state, its resonant frequency and impedance do not vary when the extension length of machining tool are fixed. Its impedance model can be taken as constant. In the machining process, the external force is transmitted to the piezoelectric vibrator through the horn and tool, and the internal material electromechanical coefficient varies with the extrusion of axial forces. Therefore, the resonant frequency, impedance, and vibration amplitude of the ultrasonic transducer affect the quality of the machining process. The external force effect can be described by a dynamics simulation model, its equivalent MSD system is shown in Figure 8a. When the ultrasonic power driver provides excitation voltage to the ultrasonic transducer by brass electrodes, the piezoelectric ceramics output the ultrasonic vibration force to push the horn structure. The initialization displacement of piezoelectric ceramics is *x*_0_(t), the piezoelectric ceramics part can be taken as one-order MSD system. *M*_p_ is the mass of piezoelectric ceramic stacks, *B*_p_ and *K*_p_ are the damping coefficient and spring rigidity of piezoelectric ceramics, respectively. Furthermore, the initialization displacement is amplified by the horn structure. The front slab, back slab, solid horn, clamping nut, collet, and machining tool are another one-order MSD system, *M*_h_, *B*_h_, and *K*_h_ are the mass, the damping coefficient and spring rigidity of MSD system. Then the vibration displacement is transferred to the workpiece. Based on Mason’s rule [16,32], the impedance model of ultrasonic transducer also can be equivalent to electrical equivalent model as shown in Figure 8b, the *L*_1_ is the dynamic inductance, *C*_1_ is the dynamic capacitance, *R*_1_ is the dynamic resistance of ultrasonic transducer, and *R*_a_ is the current sample resistance (*R*_a_ << *R*_1_).

A dynamics expression of the piezoelectric actuator can be formulated to express the displacement of the piezoelectric ceramics *x*_o_(t); the dynamics function is:(15)$$M_{p}{\ddot{x}}_{0}=B_{h}({\dot{x}}_{1}-{\dot{x}}_{0})+K_{h}(x_{1}-x_{0})-B_{p}{\dot{x}}_{0}-K_{p}x_{0}+F_{a}$$

The *x*_1_ (*t*) movement function of the machining tool tip is:
(16)$$M_{h}{\ddot{x}}_{1}=-B_{h}({\dot{x}}_{1}-{\dot{x}}_{0})-K_{h}(x_{1}-x_{0})-F_{e}$$
where *F*_e_ is the contact force, and *F*_a_ is the ultrasonic vibration force.

The magnitude displacement of horn structure is:
(17)$$G_{M}=\frac{x_{1}}{x_{0}}$$

The electromechanical impedance transfer function without external force based on Mason’s rule [16,32] is expressed as:
(18)$$G(s)=\frac{U_{i}}{I}=\frac{as^{2}+bs+1}{C_{0}s(as^{2}+bs+c+1)}$$
where $a=(G_{M}M_{h}+M_{p})/(K_{p}),b=\frac{B_{p}}{K_{p}},c=\frac{N^{2}}{G_{M}C_{0}}$.

The electrical impedance without external force in Figure 8b is expressed as:(19)$$Z_{e}(\omega )=\frac{(\omega^{2}L_{1}C_{1}-1)-j\omega C_{1}R_{1}}{\omega^{2}C_{0}C_{1}R_{1}+j[\omega^{3}L_{1}C_{1}C_{0}-\omega (C_{1}+C_{0})]}$$
where *ω* = 2*πf* is the angular frequency and *f* is natural frequency.

The electrical impedance function is expressed as:
(20)$$G(s)=\frac{U}{I}=\frac{L_{1}C_{1}s^{2}+R_{1}C_{1}s+1}{C_{0}s(L_{1}C_{1}s^{2}+R_{1}C_{1}s+\frac{C_{1}}{C_{0}}+1)}$$

It is observed that the dynamics impedance transfer function can be equivalent to the electrical impedance transfer function from Equations (18) and (20). The parameters are set as a = *L*_1_*C*_1_, b = *R*_1_C_1_, c = *C*_1_/*C*_0_.

The state space matrix of ultrasonic transducer without external force is expressed as:(21)$$\left[\begin{array}{c} \frac{du_{C_{0}}(t)}{dt}\\ \frac{di_{L_{1}}(t)}{dt}\\ \frac{du_{C_{1}}(t)}{dt} \end{array}\right]=\left[\begin{array}{ccc} -\frac{1}{C_{0}R_{a}} & -\frac{1}{C_{0}} & 0\\ \frac{1}{L_{1}} & -\frac{R_{1}}{L_{1}} & -\frac{1}{L_{1}}\\ 0 & \frac{1}{C_{1}} & 0 \end{array}\right]\left[\begin{array}{c} u_{C_{0}}(t)\\ i_{L_{1}}(t)\\ u_{C_{1}}(t) \end{array}\right]+\left[\begin{array}{c} \frac{1}{C_{0}R_{a}}\\ 0\\ 0 \end{array}\right]u_{i}(t)$$
where $u_{C_{0}}(t)$ is the voltage of the static capacitance,$i_{L_{1}}(t)$ is the current of equivalent dynamic inductance, and $u_{C_{1}}(t)$ is the voltage of equivalent dynamic capacitance.

Figure 8a shows the MSD dynamic system of ultrasonic transducer. The coupling factor is *N*. The input voltage is set to *U*_i_. The current in the closed-loop system is obtained to analyze the vibration amplitude of ultrasonic transducer. The external force is *F*_e_. The overall closed-loop transfer function of the ultrasonic transducer is shown in Figure 9.

When the machining tool impacts and separates from the processing interface, the deformation of the piezoelectric ceramic stacks is affected by the contact force. When the ultrasonic transducer is pushed by the *Z*-axial motion platform structure, the force sensor is as the contact interface to test the axial force. The impedance and phase of the ultrasonic transducer with different axial forces in frequency sweeping are measured by the impedance analyzer are shown in Figure 10. The impedance and phase parameters of ultrasonic transducer provide the data set for overall dynamic and precise impedance modeling. It is found that the resonant frequency *f*_r_ and anti-resonant frequency *f*_a_ go up with the increase of external force, the corresponding lowest impedance *Z*_r_ (resonant frequency) increases and corresponding highest impedance *Z*_a_ (anti-resonant frequency) decreases. In the machining process, it is investigated that the impedance-frequency characteristics of ultrasonic transducer drastically varies with the external force.

The resonant frequency, anti-resonant frequency, and their corresponding impedance are measured by the impedance analyzer. Therefore, the dynamic inductance *L*_1_, the dynamic capacitance *C*_1_, the dynamic resistance *R*_1_, and the static capacitance *C*_0_ are estimated by the impedance analyzer. To precisely establish the dynamic impedance transfer function, the parameter fitting method is applied in the impedance modeling. The parameters trends with different external forces are calculated and are shown in Figure 11. It is found that when the external force linearly increases, the parameter ‘a’ linearly decreases, the parameter ‘b’ linearly increases and the parameter ‘c’ decreases when external force is less than 11N and increases from 11N to 50N. The corresponding fitting functions and corresponding coefficients of determination R^2^ are listed in Table 3. The parameter fitting functions can meet the fitting accuracy of impedance transfer function of ultrasonic transducer with external force.

After the calculation and parameter fitting process, the parameters in the impedance model of the transducer are obtained to establish the accurate dynamic impedance model. The dynamic impedance transfer function with external force is expressed as
(22)$$G(s)=\frac{\left[\begin{array}{l} (3.857\times 10^{-11}-2.913\times 10^{-12}\cos(0.0173F_{e})-2.787\times 10^{-12}\sin(0.0173F_{e}))s^{2}\\ +(1.536\times 10^{-9}F_{e}+4.126\times 10^{-8})s+1 \end{array}\right]}{\left[\begin{array}{l} C_{0}s((3.857\times 10^{-11}-2.913\times 10^{-12}\cos(0.0173F_{e})-2.787\times 10^{-12}\sin(0.0173F_{e}))s^{2}\\ +(1.536\times 10^{-9}F_{e}+4.126\times 10^{-8})s\\ -4.611\times 10^{-16}\sin(F_{e}-\pi )+2.386\times 10^{-17}\times {\left(F_{e}-10\right)}^{2}+1.692\times 10^{-13}+1) \end{array}\right]}$$

To analyze the dynamic electrical response of ultrasonic transducer, the input voltage *U*_i_ is assumed as 10 × sin (ωt) V, the current response of ultrasonic transducer can be obtained in the closed-loop system (Figure 9) and Equation (21). The current responses of ultrasonic transducer are shown in Figure 12. It is observed that the peak current response of ultrasonic transducer without force factor in Equation (20) oscillates in the initialization time with different external forces, while the peak current response of ultrasonic transducer with force factor in Equation (22) rises smoothly and gets a greater value. Therefore, the dynamic impedance model is an important link of maintaining the ultrasonic transducer to work at its resonant state.

Through experiment and parameter fitting, it is found that the impedance transfer function of the transducer varies with the force. When the impedance model of the ultrasonic transducer is adopted as the control object, the variable load impedance transfer function should be considered into the resonant vibration control system.

## 6. Conclusions

In this work, an impedance model of the ultrasonic transducer in the ultrasonic machining is established. By using the electromechanical equivalent circuit method, the equivalent impedance of each component of the ultrasonic transducer is derived, and the assembled equivalent impedance model is studied. The impedance equivalent model is calculated and effectively predicts the frequency, susceptance and conductance of the ultrasonic transducer. The effects of the resonant frequency and impedance of the ultrasonic transducer with different tightening torques of clamping nut and various extension lengths of machining tool are analyzed. In the experiment, the impedance dynamic impedance model with external force is established to obtain dynamic characteristic of machining process.

## Figures and Tables

**Figure 1 sensors-19-01405-f001:**
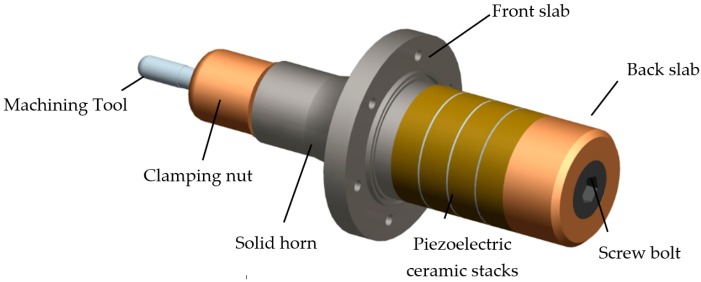
Mechanical structure of ultrasonic transducer.

**Figure 2 sensors-19-01405-f002:**
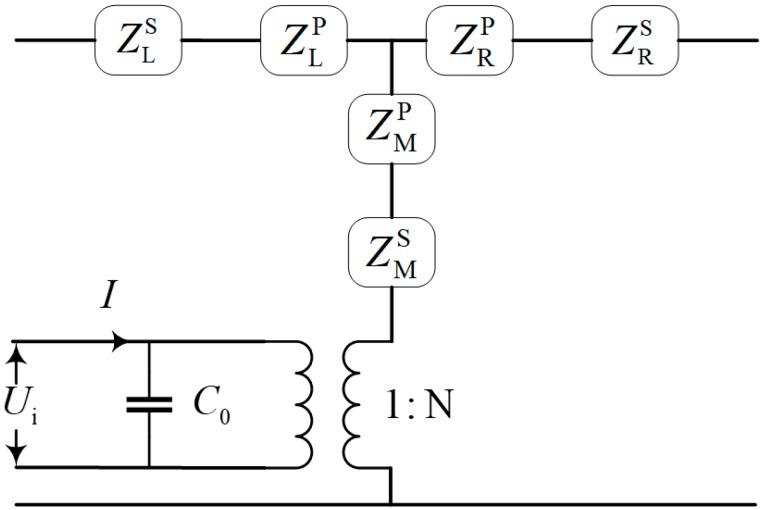
The equivalent circuit of the piezoelectric stack and screw bolt.

**Figure 3 sensors-19-01405-f003:**
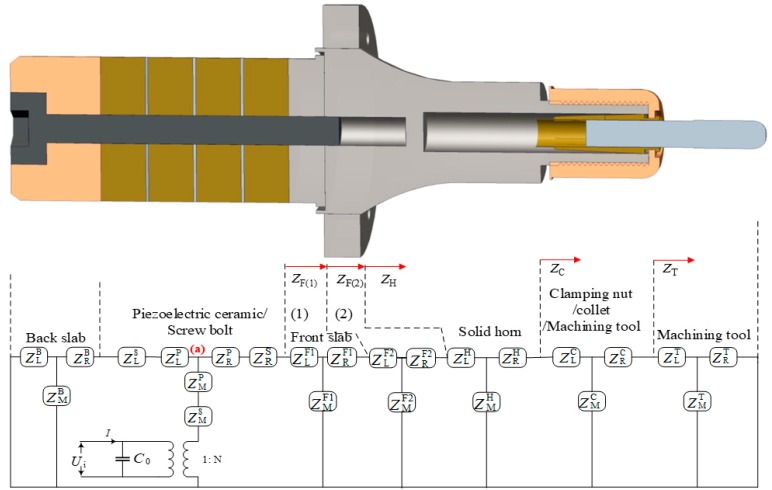
The equivalent circuit model of the ultrasonic transducer.

**Figure 4 sensors-19-01405-f004:**
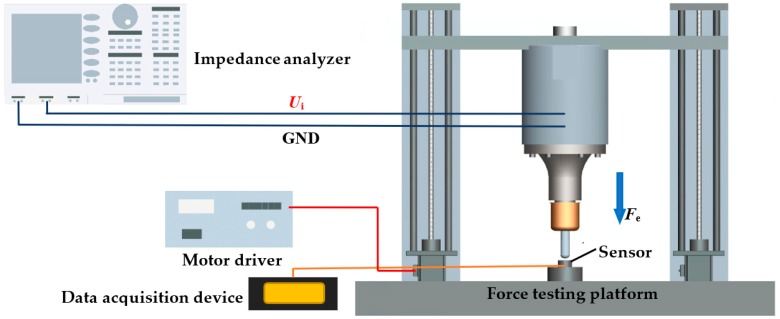
The experimental platform.

**Figure 5 sensors-19-01405-f005:**
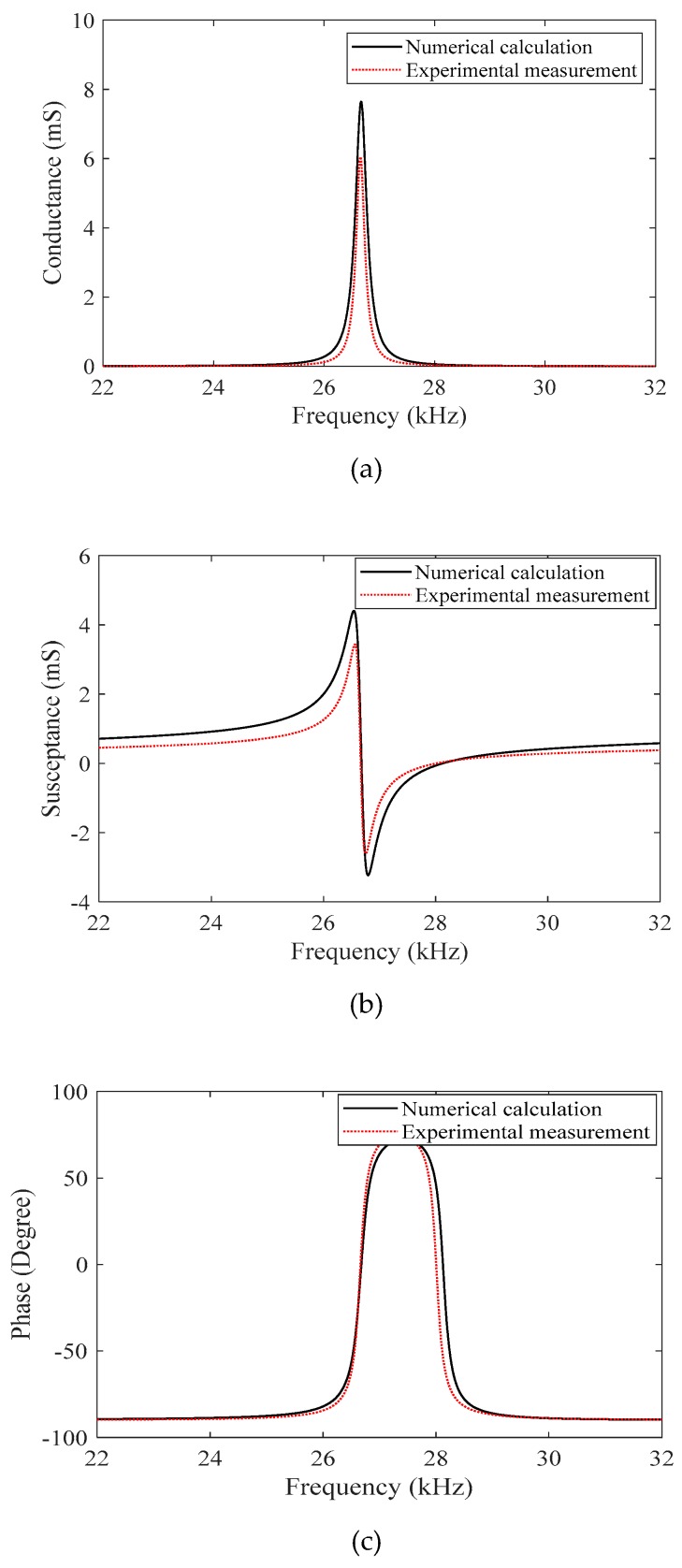
The impedance characteristics of the ultrasonic transducer. (**a**) Conductance; (**b**) Susceptance; (**c**) Phase.

**Figure 6 sensors-19-01405-f006:**
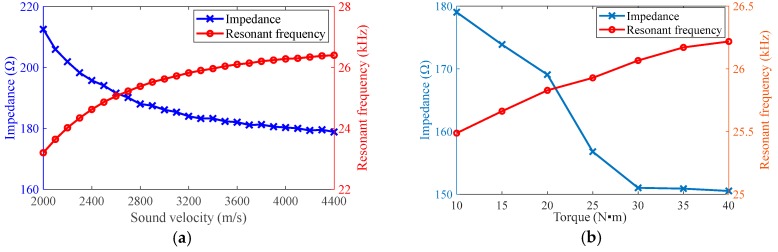
Resonant frequency and impedance with different sound velocity and tightening torque. (**a**) Theoretical calculation; (**b**) Experimental measurement.

**Figure 7 sensors-19-01405-f007:**
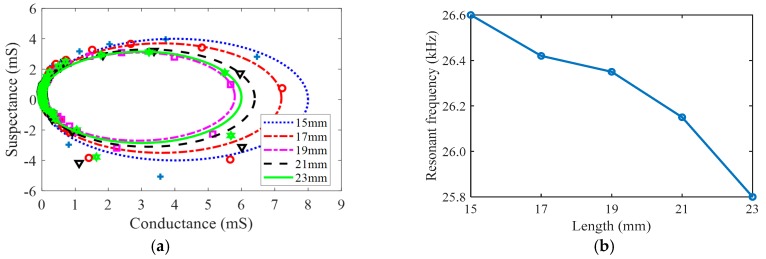
Admittance circle and resonant frequency with different lengths. (**a**) Admittance circle; (**b**) Resonant frequency.

**Figure 8 sensors-19-01405-f008:**
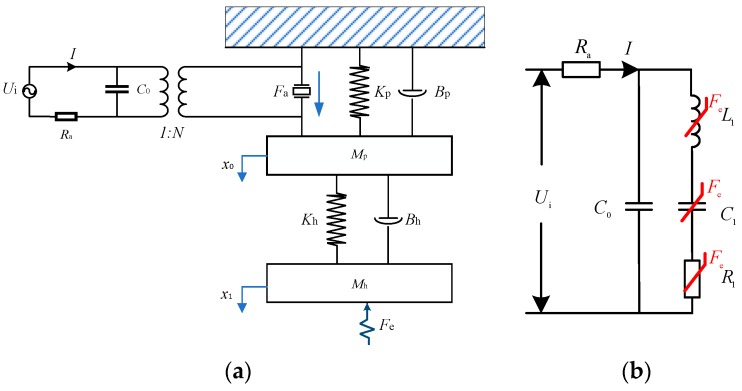
The equivalent model of ultrasonic transducer. (**a**) Dynamic MSD equivalent model; (**b**) Electrical equivalent impedance model.

**Figure 9 sensors-19-01405-f009:**
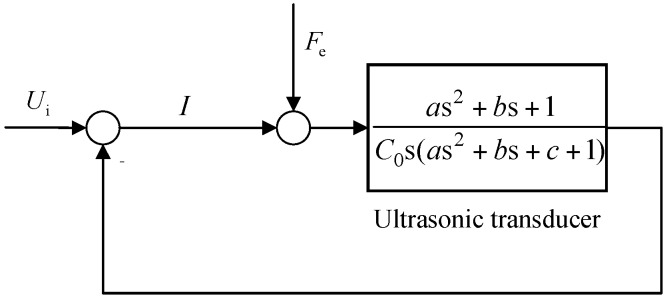
The closed-loop system of ultrasonic transducer.

**Figure 10 sensors-19-01405-f010:**
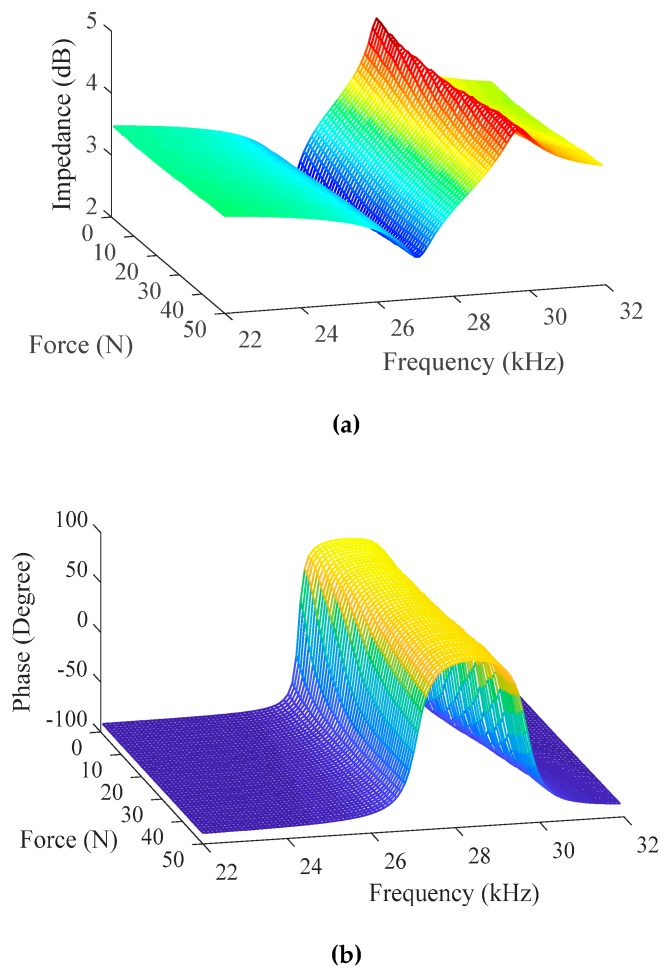
Impedance and phase with different loads. (**a**) Impedance; (**b**) Phase.

**Figure 11 sensors-19-01405-f011:**
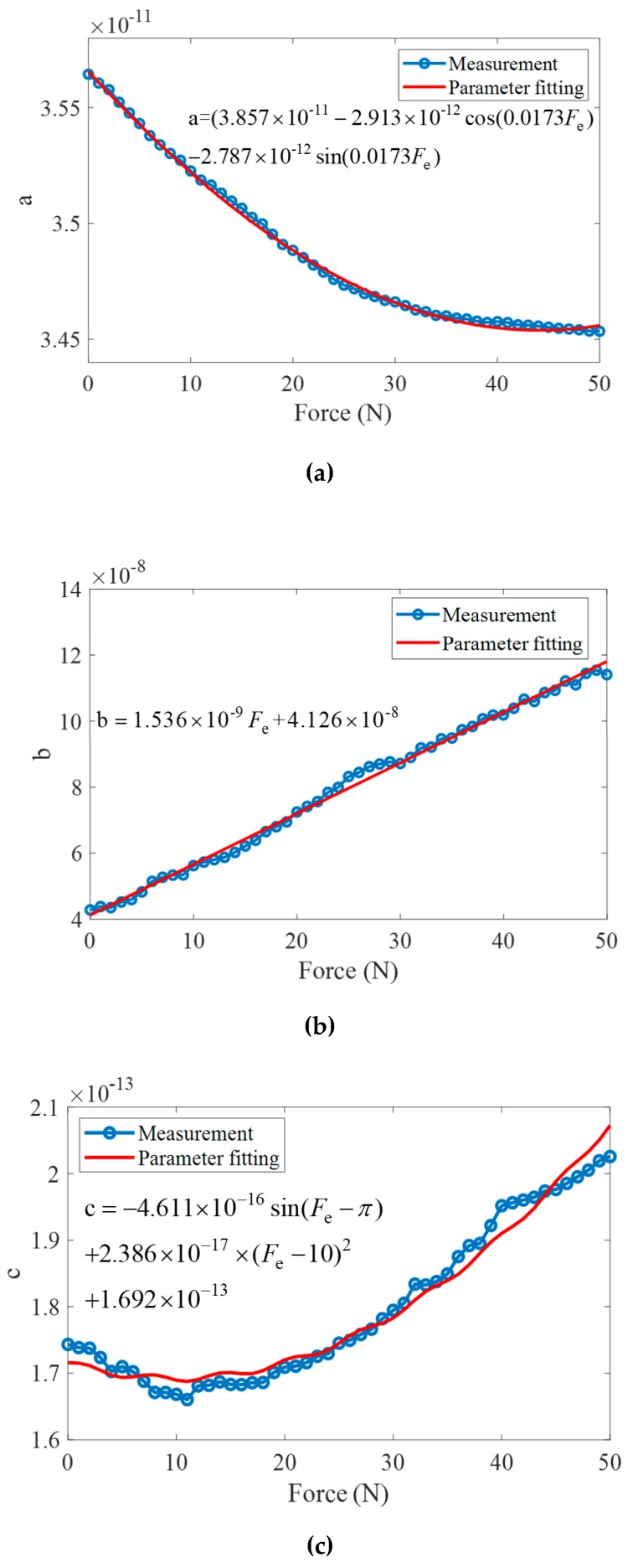
The estimated parameters of transfer function. (**a**) Parameter: a; (**b**) Parameter: b; (**c**) Parameter: c.

**Figure 12 sensors-19-01405-f012:**
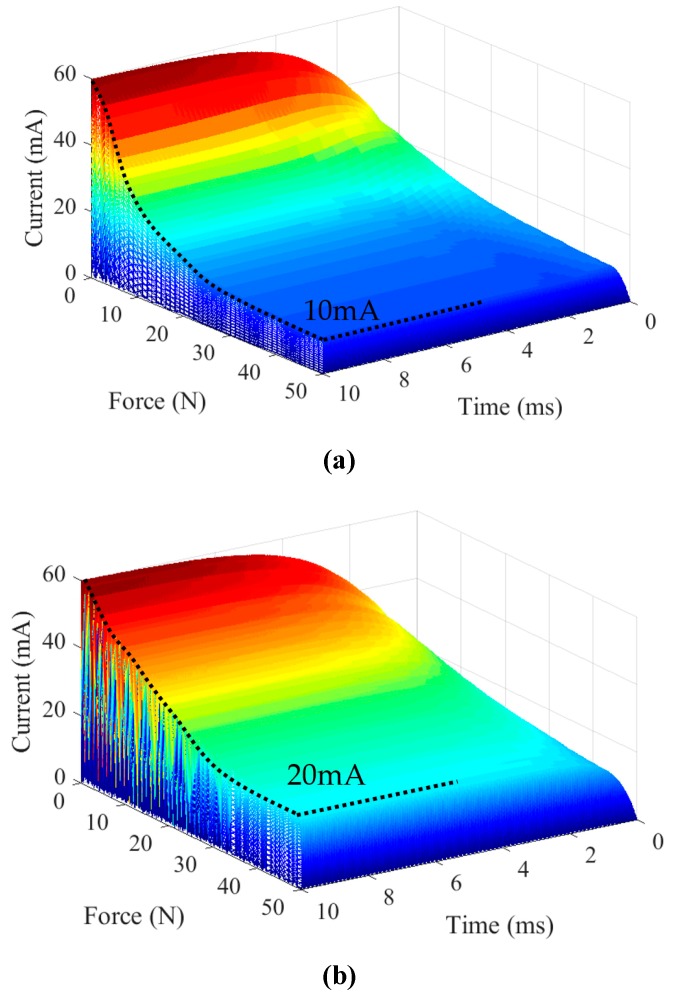
Current response of closed-loop system. (**a**) Current response without force factor; (**b**) Current response with force factor.

**Table 1 sensors-19-01405-t001:** Impedance of the horn structure in the analytical model.

Type	Shape	Parameters	Equations
**Constant horn**	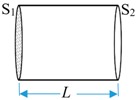	*S* = *S*_1_ = *S*_2_ *S*_1_ and *S*_2_ are the area of front and back section. *L* is the horn length.	$\begin{array}{l} Z_{L}=Z_{R}=\rho cS(\frac{1}{j\tan(\tau L)}-\frac{1}{j\sin(\tau L)})\\ Z_{M}=\frac{\rho cS}{j\sin(\tau L)} \end{array}$
**Exponential horn**	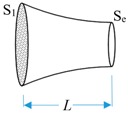	*S*_e_ = *S*_1_e^−2βL^ $\tau_{1}=\sqrt{\tau^{2}-\beta^{2}}$ *S*_1_ and *S*_e_ are the area of front and back section.	$\begin{array}{l} Z_{L}=\rho cS_{1}(\frac{\tau_{1}}{\tau }\frac{1}{j\tan(\tau_{1}L)}-\frac{\beta }{j\tau })-\frac{\tau_{1}}{\tau }\frac{\rho c\sqrt{S_{1}S_{e}}}{j\sin(\tau_{1}L)}\\ Z_{M}=\frac{\tau_{1}}{\tau }\frac{\rho c\sqrt{S_{1}S_{e}}}{j\sin(\tau_{1}L)}\\ Z_{R}=\rho cS_{e}(\frac{\tau_{1}}{\tau }\frac{1}{j\tan(\tau L)}+\frac{\beta }{j\tau })-\frac{\tau_{1}}{\tau }\frac{\rho c\sqrt{S_{1}S_{e}}}{j\sin(\tau_{1}L)} \end{array}$

**Table 2 sensors-19-01405-t002:** Properties of the ultrasonic transducer in the analytical model.

	Type	Density (kg. m^−3^)	Poisson Ratio	Dimensions		Other Parameters
				Diameter (mm)	Length (mm)	
**Piezoelectric Ring**	PZT-4	7700	0.25	Outer:32.0 Inner:13.9	6	d_33_ = 270(1 − 0.0003i) pC/N $S_{E}^{33}$ = 2 × 10^11^(1 − 0.0012i) m^2^/N
Screw bolt	Stainless steel	7930	0.28	13.5	36.5	E = 2.15 × 10^11^(1 + 0.001i) N/m^2^
Back slab				32.0	10.8	
Front slab (1)				32.0	3.5	
Front slab (2)				43.2	5.1	
Exponential horn				S_1_:28.2 S_e_:22.1	27.8	
Clamping nut				17.1	15.2	
Machining tool				6.0	(15*) 38.7	

* Extension length of machining tool.

**Table 3 sensors-19-01405-t003:** Parameter fitting Functions and R^2^.

Parameters	Type	Function	R^2^
**a**	Fourier	$F\left(x\right)=a_{0}+a_{1}\cdot \cos\left(w\cdot x\right)+b_{1}\cdot \sin\left(w\cdot x\right)$	0.9983
**b**	Polynomial	$F\left(x\right)=p_{1}\cdot x+p_{2}$	0.9952
**c**	Linear Fitting	$F\left(x\right)=\alpha \cdot \sin(x-\pi )+\beta \cdot {\left(x-10\right)}^{2}+\delta$	0.971

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
