# Peer review of "Electromechanical Dynamics Model of Ultrasonic Transducer in Ultrasonic Machining Based on Equivalent Circuit Approach"

_sensors, 2019, doi:10.3390/s19061405_

Round 1

Reviewer 1 Report

paper is ok in parts, I recommend minor changes and I hope authors will improve the article.

The paper studied the effects of the resonant frequency and impedance of the ultrasonic transducer with different tightening torques of clamping nut and various extension lengths of machining tool. But what is its practical relevance? 

I appreciate your efforts in formulating equations but at the end of the day, readers want to know  'what value does your research adds' to existing technique.  

you randomly selected a frequency range for validation. what was the basis for such selection?

your introduction is quite short, there are many several researchers in this area, need to add 10 to 15 more citations.

the number of equations can be reduced or at least try to merge some impedance equations.

mathematically It sounds ok but experimental justifications are lacking.

you state that  the impedance dynamic model is established to obtain the faster current response and greater current value with external force. however, the frequency range or impedance range is too short. reason? (need more explanations for figures 5 and 10).

Force considered is too small but still the error in prediction of impedance is large? accuracy is not so good. explain limitations?

Author Response

Thank the reviewers sincerely for the suggestions and questions.

Reviewer 2 Report

This manuscript describes the design of an ultrasonic transducer. The characteristics of the ultrasonic transducer, solid horn-type piezoelectric transducer, were analyzed using the equivalent circuit method. The effects of extension length of the machining tool and tightening torque of the clamping nut were analyzed. Overall, the characterization of the ultrasonic transducer is analyzed well, but the novelty of this article seems unclear. Hence, this manuscript should be revised to provide more detailed information.

The literature review in introduction is pretty well written. However, the novelty of this manuscript in comparison with the reviewed precedent works should be clearly described. In the introduction, the research examples of other researchers are introduced and the purpose of the paper is explained. However, what is the novelty of analyzing the tightening torque effect of the clamping nut using the equivalent circuit? It would be helpful to understand this paper if the authors emphasize the differences from previous researches more clearly.

In section 2, why should the total length of ultrasonic transducers be multiples of the wavelength? Please quote the references. And why are authors applying a 20 to 35 kHz ultrasonic energy of vibration on the transducer? Is it the desired frequency band of this transducer?

In section 3.1, the derived equations by the previous studies should be cited properly.

In figure 3, the front slabs are divided in two parts (1) and (2) in figure 3 and Table 2, but both the equivalent circuit in figure 3 and equation (13) seem to have been treated as one T-network. Is it treated as a single part or as two parts?

In Table 2, the information about the inner diameter and the number of the oelectric ring is missing. Please add it.

In figure 4, Please add the model name of the motor driver and the data acquisition device.

In figure 6, does the impedance on the vertical axis of the graph mean the magnitude of impedance at the resonant frequency? As Fasten force increases, why does the sound speed increase? 

What are the reasons for the trends of variation shown in the figures 6, 7, 10, and 12?

In figure 8, do Mh, Bh, and Kh reflect the effects of all parts such as solid horn, clamping nut, machining tool, front slab, back slab and screw bolt?

In figure 11, please quantify the difference between measurement and parameter fittings such as the R square.

Author Response

(The authors gave the same response as above.)

Reviewer 3 Report

Reviewing of the article: Micromachines - 447692

            The paper entitled “Electromechanical Dynamics Model of Ultrasonic Transducer in Ultrasonic Machining Based on Equivalent Circuit Approach” presents an original impedance model of an ultrasonic transducer devoted to ultrasonic machining. The experimental validation is achieved about the conductance, the susceptance and the phase vs the resonant frequency. Different loads with increasing tightening torques are then investigated. In a last part, the dynamics modelling of the ultrasonic transducer is considered through a Mass-Spring-Damper equivalent model and the corresponding electrical equivalent impedance model.

The proposed impedance model with the consideration of each elementary contribution is quite original and the results are interesting. The modelling of the dynamics behaviour of the ultrasonic transducer is less convincing in so far as the different parameters of the transfer function are experimentally fitted. In this case, the model cannot be used to design the transducer, because the parameters are, for example, surely dependent on geometrical dimensions.

Therefore, I consider that this paper could be accepted after minor revision.

However, even if I am not an English writing specialist but it seems to me that some forms of sentences should be rewritten.

In the following, I address the different minor corrections that I have noted:

-          L. 38: “consists the power supply …” à “consists of the power supply …

-          L. 4: “which based …” à “which is based”

-          L. 84: “The paper is organized as six sections.” à “The paper is organized in six sections.”

-          L. 87: “verified the effect …” à “verify the effect…”

-          L. 114 & L.117 & L. 122 and maybe at other locations: “stess bolt” à “screw bolt”

-          Figure 2: What represent the indexes L, M, and R of the impedances? It is  not reported in the text.

-          Equations (3) to (6) and L. 124: The impedance Z0P is referenced but not the impedance Z0S.

-          Table 1:

o   There is an error in the equation giving the impedance ZL or ZR.

o   “exponential” is split in a wrong place.

-          Figure 3: the letter “A” reported in the text is difficult to read. Maybe

-          Equation (12): The index “nut” for the impedance is never considered in the text. I suppose that the right index is “C”, which concerns the clamping nut.

-          L. 171: “as image part” à “as imaginary part”

-          Table 2: There is a typographical error for the data about the elastic properties (S33E and E): “1011” à”1011” (superscript numbers).

Figure 11: The fitting functions are quite surprising. The choice of the fitting functions is unfortunate not justified by the authors in the text of article.

Author Response

(The authors gave the same response as above.)

Round 2

Reviewer 2 Report

The revised manuscript reflects all the comments of mine. Hence, the revised manuscript is judged to be suitable for publication.